# Liquid spherical shells are a non-equilibrium steady state of active droplets

Alexander M. Bergmann[1,6], Jonathan Bauermann [2,3,6],
Giacomo Bartolucci [2,3,6], Carsten Donau[1,6], Michele Stasi[1],
Anna-Lena Holtmannspötter[1], Frank Jülicher [2,3,4], Christoph A. Weber [5] ✉ &
Job Boekhoven [1] ✉

Liquid-liquid phase separation yields spherical droplets that eventually coarsen to one large, stable droplet governed by the principle of minimal free energy. In chemically fueled phase separation, the formation of phase-separating molecules is coupled to a fuel-driven, non-equilibrium reaction cycle. It thus yields dissipative structures sustained by a continuous fuel conversion. Such dissipative structures are ubiquitous in biology but are poorly understood as they are governed by non-equilibrium thermodynamics. Here, we bridge the gap between passive, close-to-equilibrium, and active, dissipative structures with chemically fueled phase separation. We observe that spherical, active droplets can undergo a morphological transition into a liquid, spherical shell. We demonstrate that the mechanism is related to gradients of short-lived droplet material. We characterize how far out of equilibrium the spherical shell state is and the chemical power necessary to sustain it. Our work suggests alternative avenues for assembling complex stable morphologies, which might already be exploited to form membraneless organelles by cells.

Spontaneous structure formation through self-assembly and phase separation is essential in biology and engineering[1]. These structures form through free energy minimization, leading to in- or close-to-equilibrium morphologies. Typical examples are structures formed by amphiphiles[2–4], nanoparticles[5], liquid crystals[6], peptides[7–9], and the demixing of immiscible liquids[10], which find application in healthcare[11], optoelectronics[12], and others. In contrast, dissipative structures can only form when a system is forcefully kept from reaching a free energy minimum by a continuous energy supply[13–16]. Active droplets exemplify a dissipative structure since their maintenance requires a continuous supply of free energy and matter[17–23]. These structures are ubiquitous in biology[24] but remain poorly understood[25], and synthetic

models are rare[26]. In contrast, in- or close-to-equilibrium droplets are well-understood[27–32]. Such droplets form via phase separation in mixtures of immiscible liquids. They tend towards a spherical shape, corresponding to minimal interfacial surface energy. Minimizing this energy also drives coarsening through fusion and Ostwald ripening, yielding a single droplet (Fig. 1a).

Active droplets are a class of dissipative structures whose properties are governed by chemical reactions and diffusive fluxes. Theoretical studies on these structures show behaviors that result from their non-equilibrium nature, which includes inhibiting Ostwald ripening and thereby controlling their size[33,34] and spontaneous self-division[35,36]. These behaviors require a non-equilibrium steady state

[1]School of Natural Sciences, Department of Chemistry, Technical University of Munich, Lichtenbergstrasse 4, 85748 Garching, Germany. [2]Max Planck Institute for the Physics of Complex Systems, Nöthnitzer Strasse 38, 01187 Dresden, Germany. [3]Center for Systems Biology Dresden, Pfotenhauerstrasse 108, 01307 Dresden, Germany. [4]Cluster of Excellence Physics of Life, Technical University of Dresden, 01307 Dresden, Germany. [5]Faculty of Mathematics, Natural Sciences, and Materials Engineering: Institute of Physics, University of Augsburg, Universitätsstrasse 1, 86159 Augsburg, Germany. [6]These authors contributed equally: Alexander M. Bergmann, Jonathan Bauermann, Giacomo Bartolucci, Carsten Donau. ✉e-mail: christoph.weber@physik.uni-augsburg.de; job.boekhoven@tum.de

**a** Phase separation in equilibrium
(passive droplets)

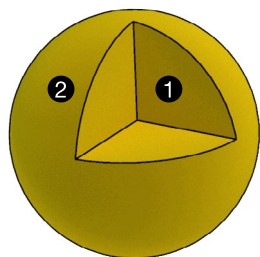

**b** Chemically fueled phase separation
(active droplets)

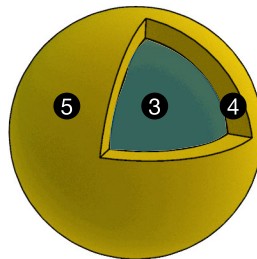

**Fig. 1 | Passive versus active droplets. a** Phase separation close to thermodynamic equilibrium leads to spherical droplets. 1) A liquid droplet coexists with its surrounding dilute phase. 2) Interfacial surface tension decreases the surface area: droplets are spherical and coarsen. **b** An alternative morphology is described in this work: a liquid, spherical shell. 3) A core consisting of a liquid phase similar to the surrounding dilute phase. 4) A shell of liquid droplet material. 5) The unfavorable high-surface-area shell is sustained by the conversion of chemical energy.

which has been hard to achieve under experimental conditions[37,38]. Synthetic analogs are scarce to test these behaviors, and methods to continuously fuel these droplets do not exist.

In this work, we describe how continuously fueled phase separation yields a dissipative structure: a thin, spherical shell of phase-separated liquid (Fig. 1b). This is intriguing since the additional interface and, thus, the larger surface area compared to volume makes such spherical shells thermodynamically unstable. The energy supply to sustain the thermodynamically unstable state comes from continuously converting fuel to waste, leading to diffusive fluxes that stabilize the spherical shell. We show that this non-equilibrium spherical shell state forms due to an instability of the active droplet´s core and that the shell interfaces act as a pump for fuel-activated chemical components.

## Results
### Fuel-driven complex coacervation

In our chemically fueled droplets, DIC (N,N′-diisopropylcarbodiimide) is the high energy molecule (fuel) that reacts with the C-terminal aspartic acid of a peptide (Ac-F(RG)₃D-OH, precursor, Fig. 2a). It converts the precursor into its corresponding cyclic anhydride (product, Fig. 2a). That product has a half-life of 58 s ($k_{deact.}$= 0.012 ± 0.009 s⁻¹, Supplementary Fig. 1a–c, Supplementary Table 1) before it is deactivated through hydrolysis. Upon activation, the peptide turns from zwitterionic to cationic with an overall charge of +3 ($k_{act.}$ = 0.17 ± 0.008 M⁻¹ s⁻¹, Supplementary Fig. 1a–c, Supplementary Table 1). In their short lifetime, the cationic, activated peptide can bind the polyanion polystyrene sulfonate (pSS, 400 monomer repeat units, 75 kg mol⁻¹). The non-activated, zwitterionic precursor interacts weakly with pSS ($K_{D, prec.}$ of 104.5 ± 11.4 μM, Supplementary Table 2, Supplementary Fig. 1d). In contrast, the product interacts strongly with pSS ($K_{D, prod.}$ of 2.4 ± 1.2 μM, Supplementary Table 2, Supplementary Fig. 1e and Supplementary Discussion 1). Thus, chemical fuel creates a population of short-lived activated building blocks that can interact with polyanions and phase separate into complex coacervate droplets.

### Size-dependent spherical shell formation upon dissolution

When we fueled a solution of 22 mM peptide and 12 mM pSS (the concentration is expressed in repeat units) in 200 mM MES buffer with a batch of 20 mM fuel, we found that the sample turned turbid (Supplementary Fig. 2a) due to the formation of micron-sized coacervate-based droplets as evidenced by confocal microscopy (Fig. 2b). These droplets fused, and, when the emulsion has depleted most of its fuel, the droplets decayed—a clear solution without droplets was obtained (Supplementary Fig. 2a). In the dissolution process, after around 7 min, when the fuel ran low, we found that the droplets transitioned from a homogenous droplet to a spherical shell (Fig. 2b, Supplementary

Movie 1), a process referred to as vacuolization[39–41] which is also reminiscent of bubbly phase separation in active fluids[42]. Moreover, we found that the larger the droplet, the sooner the droplet became unstable and formed a spherical shell (Fig. 2c, d). During the dissolution, multiple vacuoles could form within one coacervate-based droplet, which eventually fused into one. Additionally, the vacuole interior could also be expelled if the surrounding coacervate phase was too thin, leading to the collapse of the spherical shell into a droplet (Fig. 2e). This process was observed repeatedly until the coacervate-based droplets fully dissolved.

If the solution was fueled with only 10 mM DIC, the droplets stayed smaller and did not form shells during their dissolution (Supplementary Fig. 2b, c, Supplementary Movie 2). Vacuole formation has previously been shown as a dissolution pathway of coacervate-based droplets[18,43,44]. Furthermore, the expulsion of the vacuole interior has even been demonstrated to propel the movement of the coacervate-based droplets[45]. In our case, we hypothesize that these spherical shells form because when fuel runs low, the influx of new droplet material decreases, and the deactivation of the peptide in the core outcompetes the influx of new material. Thus, gradients of product material could form within the droplet, leading to the dissolution of the droplet's core. A larger droplet would have a steeper gradient and thus dissolve sooner. The initial formation of multiple vacuoles in one droplet instead of a single central vacuole has been reported previously due to temperature changes that destabilized the droplets and is dependent on how rapidly they are destabilized[46]. In our case, it is most likely due to the rapid decay of fuel and product. This mechanism suggests that spherical shells could form and be sustained in a steady state when fuel is provided continuously to very large droplets.

### Continuously fueled active droplets in microreactors

Continuously fueling soft matter to yield sustained non-equilibrium steady states is challenging because these solutions cannot be stirred without affecting the assemblies. Moreover, waste accumulation often inhibits the self-assembling process[38,47]. Therefore, we developed aqueous microreactors into which fuel continuously diffuses from a reservoir (Fig. 3a, b). We made these aqueous microreactors by preparing a stable emulsion of water droplets in a fluorinated oil phase. The water droplets (microreactors) contained the peptide, buffer, and polyanion, i.e., all reagents except the fuel. Notably, we reduced both the precursor (10 mM) and pSS (5 mM) concentrations in comparison to the batch-fueling experiments to minimize fuel gradients within the reactors. The fluorinated oil contained the fuel, which diffused into the microreactors at the oil-water interface until a steady state of fuel and product was reached (Fig. 3a and Supplementary Fig. 3a, b).

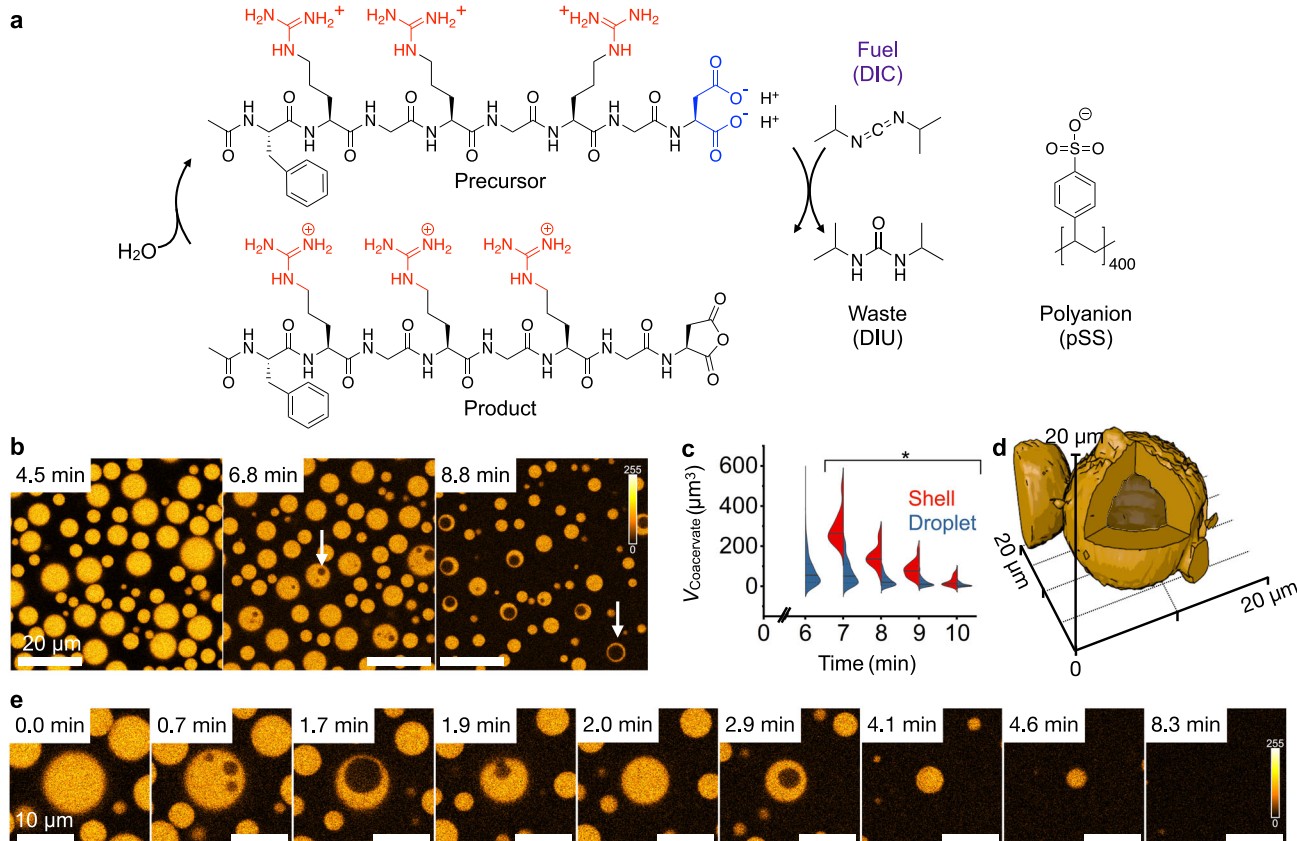

**Fig. 2 | Chemically fueled droplets under batch-fueling. a** Chemical structures of the components that form chemically fueled droplets and liquid shells. **b** Confocal microscopy of a solution of 22 mM precursor, 12 mM pSS, and 0.1 μM sulforhoda-mine B in 200 mM MES buffer at pH 5.3 fueled with 20 mM DIC. After 7 min, the active droplets became unstable and swelled to form a spherical shell (white arrows). The scale bar of all images represents 20 μm. The color scale is given next to the images. All experiments were performed in triplicate ($N = 3$). **c** The volume distribution of active droplets (blue) and active liquid shells (red) is shown over time. The solid line represents the median, and the dotted lines represent the upper and lower quartiles. Larger active droplets formed spherical shells earlier than smaller active droplets. * $P$-values < 0.05 between the shell and droplet population for each time point. **d** 3D reconstruction from confocal microscopy data shows the spherical nature of the shell. Conditions were 10 mM precursor, 5 mM pSS, and 0.1 μM sulforhodamine B in 200 mM MES buffer at pH 5.3 fueled with an excess of DIC (5 μL) on top of 20 μL of the sample. The diffusion of excess fuel into the sample leads to the formation of bigger droplets that are longer lived. The Z-stack for the 3D reconstruction is imaged two hours after adding fuel. **e** The shrinkage and dissolution of a representative droplet from **b**. The scale bar of all images represents 10 μm. The color scale is given next to the images. Source data are provided as a Source Data file.

Additionally, the waste of the reaction cycle (DIU, diisopropylurea) partitioned preferentially outside the microreactors and crystallized in the fluorinated oil phase (Supplementary Fig. 3a, c). For the experimental conditions of Fig. 3 (0.5 M DIC in the oil phase), a concentration of 8.4 mM DIC in the aqueous phase was measured (Supplementary Fig. 3d). Moreover, the concentration of fuel in the fluorinated oil also controlled the fuel concentration inside the microreactor (Supplementary Fig. 3d). Thus, this continuous fueling method allows us to continuously fuel microreactors at various steady states and simultaneously avoid the challenge of accumulating waste. Z-stacks of the microreactors can be acquired with a confocal microscope. When projected onto one plane, each coacervate droplet can be counted, and its volume measured. Finally, with our method, many microreactors of volumes ranging from thousands to millions of μm³ are made in one experiment.

**Size-dependent transition into active spherical shells**
Within seconds after preparing the microreactors, coacervate droplets nucleated and grew through fusion (Fig. 3a). After about 10 min, the total volume of all combined droplets within one microreactor reached its maximum. It stayed constant throughout the experiment, indicating a steady state in activation and deactivation was reached (Supplementary Fig. 3e, f, Supplementary Movie 3). We investigated

the influence of the steady-state fuel concentrations on the total amount of coacervate-based droplets in each container (Supplementary Fig. 3g). This revealed that the fuel concentration within the reactors controlled the total amount of coacervate-based droplets within each reactor resulting in bigger coacervate-based droplets the higher the fuel concentration was set. Within 30 min, all droplets fused until one large active droplet remained. We identified two possible behaviors of the active droplets: in small microreactors, the active droplet sunk to the bottom of the reactor, slightly wetted the reactor floor and remained stable for the entire observation time (Fig. 3a, Supplementary Figs. 4a and 5a, b, Supplementary Movie 4).

In large microreactors, the active droplet was also larger (Fig. 3b). These droplets sunk and slightly wetted the reactor floor. However, after about 2 h, its core became unstable. The droplet transitioned and left behind a shell of droplet material homogenous in thickness that was stable for several hours (Supplementary Fig. 5c–f, Supplementary Movie 5). As the original droplet had wetted the floor, we did not obtain a spherical shell as for batch-fueling experiments. Instead, we observed a half dome of phase-separated material (Fig. 3c, d). If we performed the same experiments with passive droplets, we did not observe the formation of any spherical shells (Supplementary Fig. 4b–g). We measured the product's diffusion coefficient within the active droplets and the active spherical shells, $D_{prod.}$, by fluorescence recovery after

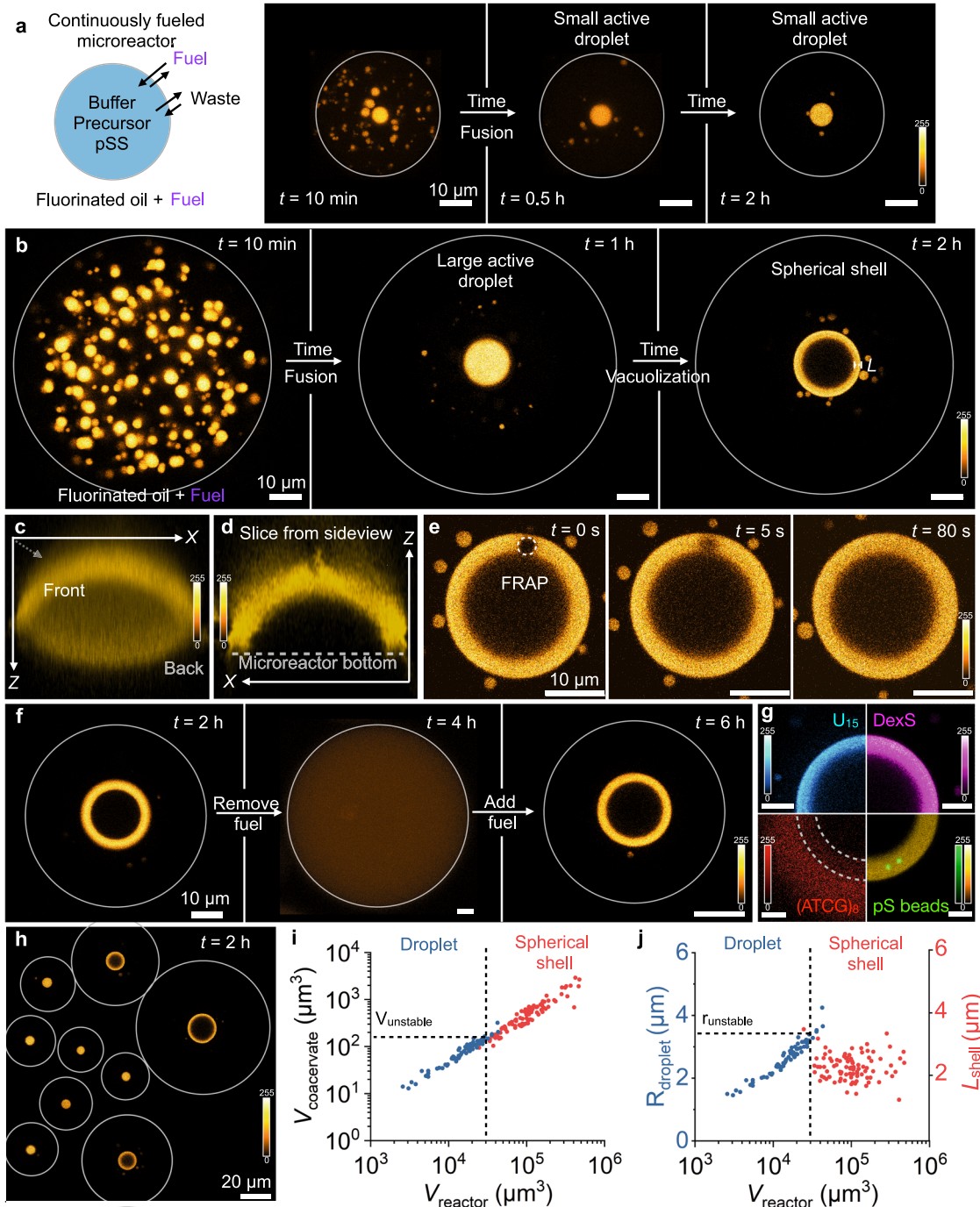

photobleaching (FRAP) to be 0.03 μm² s⁻¹ (Fig. 3e and Supplementary Fig. 6b, e, Supplementary Table 3). Additional FRAP experiments on the precursor, and the polyanion (Supplementary Fig. 6a, c, d, f), confirmed the shell was liquid and not in a dynamically arrested state[48], which was further corroborated when we exchanged the fluorinated oil with fluorinated oil without fuel. The spherical shell rapidly dissolved until all droplet material had decayed (Fig. 3f). When the oil was exchanged with DIC-loaded oil, droplets reappeared that fused. Still, due to extensive wetting of the newly formed active droplets, only a small fraction of the microreactors yielded spherical shells.

Next, we tested the permeability and partitioning of the shell by adding fluorescently labeled anionic molecules and nanoparticles: a U₁₅ RNA oligomer, dextran sulfate, carboxy-terminated polystyrene particles, and a DNA oligomer (ATCG)₈ (Fig. 3g). All partitioned well into the shell except for (ATCG)₈ which showed a lower fluorescence in

the shell than in the dilute phase. Additionally, both U₁₅ and dextran sulfate showed similar fluorescence within the interior of the spherical shell compared to the dilute phase outside (Supplementary Fig. 7a, b), whereas (ATCG)₈ was excluded from the interior of the spherical shells (Supplementary Fig. 7c). This suggests that the shell acts as a permeable membrane to molecules that partition well, but not ones that hardly partition or large particles that do not leave the spherical shell[49]. The zeta potential of static droplets was determined to be −46.0 ± 2.5 mV, which aligns with a generally observed partitioning behavior of coacervate-based droplets. Multivalent electrostatic interactions are one of the main driving forces for complex coacervation resulting in a high partitioning of the U₁₅ RNA oligomer and dextran sulfate. However, also cationic or less charged molecules like Rhodamine 6 G, Hoechst 33342, and Nile red are partitioned into the shells (Supplementary Fig. 7f–h)[50]. The DNA oligomer (ATCG)₈ hardly

**Fig. 3 | Spherical shells are a stable, non-equilibrium state. a, b** Experimental setup to form microreactors that continuously fuel active droplets. Surfactant-stabilized microfluidic droplets (microreactors, grey outline) containing 10 mM precursor, 5 mM pSS, and 0.1 μM sulforhodamine B in 200 mM MES buffer at pH 5.3 were embedded in a fluorinated oil which contained 0.5 M DIC (fuel) to sustain the microreactor at a fuel concentration of 8.4 mM. Time-lapsed micrographs of a small (**a**) and large (**b**) microreactor in a steady state. The grey circle represents the size of the microreactor. The images for 10 min and 0.5 h in **a** and the image for 10 min in **b** are Z-projections of the microreactor. All other images are from one Z-plane. The scale bar of all images represents 10 μm. The color scale is given next to the images. **c, d** A 3D projection of a spherical shell reconstructed from Z-stack imaging. A 3D projection of the average pixel intensity (**c**) and a slice in the XZ-plane through the middle of the spherical shell (**d**) is shown. **e** A FRAP study of the spherical shell demonstrates that the shell is liquid and dynamic. The scale bar of all images represents 10 μm. The color scale is given next to the images. **f** The microreactor was fueled with 0.5 M DIC in the oil phase. After the formation of spherical shells (2 h), the oil phase was replaced with oil containing no DIC. After a homogenous solution was obtained (4 h), the oil was replaced with 0.5 M DIC-containing oil. The grey circle represents the size of the microreactor. The scale bar of all images represents 10 μm. The color scale is given next to the images. **g** Partitioning of different fluorescently labeled molecules into the spherical shell. The dotted line represents the outlines of the spherical shell. The scale bar of all images represents 5 μm. The color scale is given next to the images. **h** A macroscopic view of multiple microreactors shows that large reactors formed spherical shells while small microreactors contained droplets. The center Z-plane of each of the individual droplets and shells is projected. The grey circles represent the individual micro-fluidic reactors. The color scale is given next to the images. **i** The volume of the total coacervate material is shown for every individual microreactor that contained an active droplet (blue) or an active shell (red). Above a critical reactor volume, active droplets with a volume bigger than $V_{unstable}$ transformed into spherical shells. **j** The radius of the active droplet (blue) and the shell thickness $L$ of the active shell (red) for every individual microreactor is shown. Spherical shells had a similar thickness $L_{shell}$, which was within the range of $r_{unstable}$. All experiments were performed in triplicate ($N = 3$). Source data are provided as a Source Data file.

showed any partitioning despite being negatively charged. We attribute the difference to the self-complementarity of (ATCG)$_8$ and its resulting double-stranded nature, which was shown to hinder partitioning into membraneless organelles[51].

We quantified the relationship between the reactor volume and droplet behavior. We first confirmed that the volume of the droplet material scales linearly with the reactor volume (Fig. 3h, i). Simply put, a larger microreactor contains more precursor molecules and produces more product molecules resulting in larger coacervate-based droplets. This dataset revealed that a shell could be expected when the microreactor was larger than roughly 30,000 μm$^3$ (Fig. 3i). This microreactor volume corresponds to a coacervate-based droplet volume of 150 μm$^3$. In other words, shells were formed when the microreactor was sufficiently large to form coacervate-based droplets larger than 150 μm$^3$ corresponding to the threshold radius $r_{unstable} = 3.3$ μm (Fig. 3j). Additionally, we found that the thickness of the spherical shells $L_{shell}$ was relatively constant ($L_{shell,exp.} = 2.4 \pm 0.4$ μm) over an extensive range of total phase-separated material ($V_{coacervate,\,shell} = 150$ to $3000$ μm$^3$). We further tested the robustness of the observed shell formation. We found that shells were formed for varying pSS and precursor concentrations and different pSS chain lengths. However, changing the polyanion for polyvinyl sulfonate or polyuridylic acid did not lead to the formation of shells (Supplementary Discussion 3).

## Mechanism for spherical shell formation

Different mechanisms have been reported for forming morphologies similar to the spherical shells observed in this work. Stable shell-like morphologies at thermodynamic equilibrium have previously been reported due to the formation of multiphase compartmentalization[52–54]. In these cases, a third coexisting phase in the droplet's core stabilizes the outer liquid shell-like phase. In our case, the transition into a spherical shell is not due to the formation of an additional coexisting dense phase in the core of the droplet but there is an additional dilute domain with a composition similar to the surrounding dilute phase, i.e., all the droplet-forming materials are depleted from the core (Supplementary Fig. 7d, e). This hypothesis is supported by the phase diagram associated with the corresponding mixture with the stabilized product (product*), which does not exhibit three-phase coexistence (Supplementary Fig. 9a, b). Another possibility is the formation of vesicular structures due to surface-active building blocks supplied[55,56] or formed in situ through, e.g., pH change[57] or charge mismatch[49]. However, the formation of new surface-active building blocks can be excluded due to the absence of the spherical shell transition in the compositionally equivalent reactors containing droplets with $r < r_{unstable}$ and the formation of stable passive droplets of similar compositions (Supplementary Fig. 4d–i). The transition of droplets into metastable shells has been reported as a transient state due to underlying reentrant phase

transitions[43], degradation of the building blocks[44], dynamically arrested states due to a change in composition[48], after deep temperature quenches[46], or by applying an electric field[41]. However, these mechanisms rely on changes in external factors or chemical composition, and the resulting non-equilibrium morphologies cannot be sustained in time. In contrast, spherical shells in our system are stationary, and their morphology can be sustained. Therefore, neither of these mechanisms explains the formation of our observed spherical shells.

To better understand the mechanism of the spherical shell formation, we measured the product's and fuel's partitioning coefficients in the coacervate-based droplets with a spin-down assay. We found that the fuel only weakly partitions inside the droplets ($K_{fuel} = 1.4 \pm 1.7$, Supplementary Table 4, Supplementary Discussion 2). Thus, most of the fuel remains outside the droplets. In contrast, the product partitions strongly inside the droplets ($K_{product} = 3,360 \pm 1,645$, Supplementary Table 5). We conclude that activation predominantly occurs outside, whereas deactivation occurs inside the droplets. The spatial separation of these reactions leads to a diffusion gradient of product that builds up from the outside of the droplet towards its core (Fig. 4a). In other words, the greater the radius of the active droplet, the more its core is depleted from the product. It thus destabilizes, leading to the core's instability and the transition into a spherical shell (Fig. 4b, c and Supplementary Fig. 7e, Supplementary Movie 6).

## Quantitative verification of spherical shell formation

To quantitatively verify that hypothesis, we developed a non-equilibrium, thermodynamics-based model that describes a droplet in the center of a spherical reactor with the fuel and waste concentrations maintained at the reactor's boundary. We used experimentally determined steady-state concentrations, rate constants, diffusion, and partitioning coefficients (Supplementary Tables 1–6). To provide a theory with minimal physiochemical ingredients, we do not describe the polyanion and the wetting of the spherical shell on the microreactor wall. We experimentally determined the concentrations of the three components in the phase diagram (see Supplementary Discussion 2), which was then fitted by a mean-field Flory Huggins model showing good agreement (Supplementary Fig. 9a, b).

We used our model to calculate the product's concentration profile from a droplet core to its boundary (Fig. 4d). We used a droplet with a 5 μm radius which is greater than $r_{unstable}$ (Fig. 3j). The concentration decreases from 1200 mM at the droplet interface to 450 mM at the core, which is depleted of droplet material because of the deactivation. The core's concentration is far below the spinodal decomposition concentration. Thus, the droplet core cannot be stable, i.e., a new domain more similar to the outside nucleates at the core and

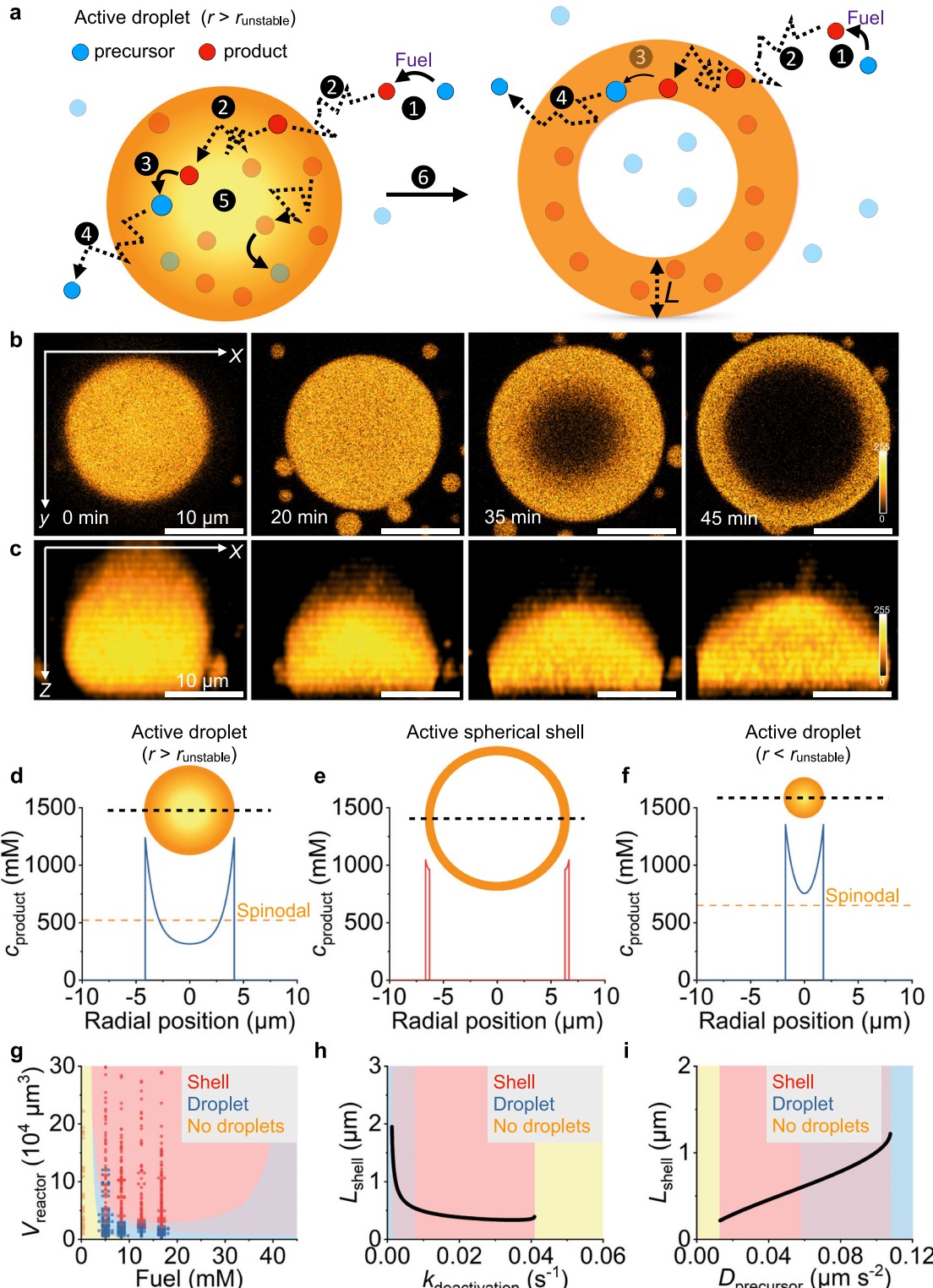

**Fig. 4 | The mechanism of spherical shell formation. a** Schematic representation of the mechanism of spherical shell formation. 1) Fuel-driven activation outside of the droplet. 2) The product diffuses into the droplet. 3) Deactivation inside of the droplet. 4) Precursor diffuses out of the droplet. 5) The droplet's core is depleted of product. 6) The droplet transitions to a spherical shell. **b, c** Confocal micrograph timelapse series of an active droplet with a critical radius larger than $r_{unstable}$. Over 20 min, the droplet wetted the microreactor's bottom and transitioned into a spherical shell. The images in **b** show the $XY$-plane close to the bottom of the reactor. The images in **c** show the $XZ$-projection. The scale bar of all images represents 10 µm. The color scale is given next to the images. Concentration profiles of a large chemically fueled droplet with $r > r_{unstable}$ (**d**) before transitioning into a spherical shell, a spherical shell (**e**), and a small chemically fueled droplet with $r < r_{unstable}$ (**f**). The insets show a scheme of the droplet and spherical shell. **g** The system's behavior as a function of steady-state concentration fuel and reactor volume. The shaded areas represent the stable state calculated by the model. The red-blue shaded area represents the coexistence of stable droplets and shells. The markers show the phase-separated state of the experimental data. Overlapping data points are shown with an offset. All experiments were performed in triplicate ($N = 3$). The shell thickness $L_{shell}$ for a microreactor with a radius of 25 µm as a function of varying deactivation rate constants (**h**) or precursor diffusion constants (**i**). Source data are provided as a Source Data file.

grows, leaving behind a spherical shell of droplet material (Fig. 4e). A droplet with a radius smaller than $r_{unstable}$ has a core in which the product concentration remains above the spinodal decomposition concentration and is thus stable (Fig. 4f). We found that the model predicted $r_{unstable}$ to be 2 μm, i.e., close to the experimentally determined 3.3 μm.

Further calculations created a diagram that predicts a droplet's behavior based on its microreactor size and fuel concentration (Fig. 4g). At low steady-state fuel concentrations, insufficient droplet material is activated to form a stable droplet. Above 5 mM of fuel, phase separation is observed, and a stable droplet or a spherical shell is found depending on the microreactor size. Large reactors produced stable droplets beyond a steady-state fuel concentration of 20 mM. We explain this because as the fuel concentration increases, its concentration inside the droplets increases. The resulting peptide re-activation inside the droplet weakens the concentration gradient of the product inside (Supplementary Fig. 10a). According to our model, the non-equilibrium steady states of an active spherical shell and an active droplet can coexist (blue-red shaded area). Some of these theoretical calculations could be experimentally verified. We changed the fluorinated oil's fuel concentration to tune the microreactors' fuel concentration between 0.4 and 16.8 mM (Fig. 4g, Supplementary Figs. 3d and 8) to find agreement with the predicted microreactor sizes for the formation of spherical shells, droplets, and the homogenous phase. Fuel concentrations above 16.8 mM could not be tested experimentally because the necessary high fuel concentrations in the oil phase did not yield stable reactors.

In line with the experiments, the model also predicted that the shell thickness (L) is relatively constant with varying reactor sizes, despite decreasing slightly towards bigger reactor volumes (Supplementary Fig. 10b, c). However, the absolute value is lower than measured ($L_{theory} \approx 0.3–0.6$ μm). Using the model, we calculated that $L$ initially decreases rapidly with an increasing deactivation rate constant ($k_{deact.}$) but is then quite stable over a broad range of $k_{deact.}$. However, no droplets were found if the product is too short-lived (Fig. 4h). In contrast, increasing the product's diffusion constant made the shell thicker, i.e., the activated product can diffuse further in its short lifetime leading to a weaker gradient. Yet, a too-fast diffusing product led to no droplets (Fig. 4i). We attempted to experimentally verify the dependence of the shell formation on the alteration of the deactivation rate and diffusion constant by a change in the external conditions. However, due to the temperature and pH dependence of the reaction rate constants, the diffusion, and the phase diagram of our system, no clear conclusion could be drawn about the effect of either of these parameters individually (Supplementary Discussion 4).

### Free energy and chemical work to sustain spherical shells
Finally, we calculated the free energy and the chemical work needed to produce spherical shells (see Supplementary Methods). The free energy difference between a spherical shell and a droplet of identical volume is reflected in the additional interface. For an assumed surface tension (γ) of 75 μN m⁻¹, the additional free energy cost ($F_{surface}$) of a spherical shell with an inner radius ($R_{in}$) of 2 μm compared to a droplet equals 4 fJ. Based on the equilibrium constant of the acid-anhydride equilibrium[58], we calculate a free energy difference between the precursor and the product of about 10 $k_B T$. Combined with the steady-state concentration of the product, we calculated that the system is chemically away from equilibrium by 80 nJ ($F_{act} = 80$ nJ). Therefore, the additional surface energy is negligible compared to the free energy needed for activating the precursor to the product to sustain the spherical shell. We calculated the number of chemical cycles to sustain the steady state to be $6 \cdot 10^4$ μm⁻³ s⁻¹. Combined with the free energy difference of the cycle, the total free energy turned over per time and volume is $J_{tot} = 0.275$ W L⁻¹. Simply put, this power is supplied to

maintain the product population in the activated state. At 0.275 W L⁻¹, the power consumption is smaller but comparable to that of living cells at 1 W L⁻¹. One reason is that the concentration of fuel (ATP in cells) and the energy it liberates upon hydrolysis are in the same range as our synthetic system[59]. Interestingly, reminiscent morphologies of spherical shells have been observed in various cell types[39,60].

Nevertheless, what fraction of this power maintains the spherical shell morphology? The spherical shell results from activation dominating deactivation outside, while net deactivation is inside. This imbalance of chemical rates gives rise to a net diffusive influx of product and efflux of the precursor at the shell interfaces, $J_{int}$. The flux of the product multiplied by the activation energy of 10 $k_B T$ corresponds to the power to keep the shell interface steady and prevent the relaxation to the active droplet or even the homogenous state. Thus, the imbalance of chemical rates between the two coexisting phases gives rise to an energy pump. We find that the power of this pump is about $J_{int} = 0.198$ W L⁻¹ (see Supplementary Methods). The ratio between the pump power, $J_{int}$, and the total supplied chemical power, $J_{tot}$, is approximately 0.7 and can be thought of as a measure of the pump efficiency. $J_{int}/J_{tot}$ is lower than 1 because the deactivation outside and the re-activation inside the shell do not contribute to the flux through the interface.

## Discussion
Our microreactors offer a method to fuel self-assembling systems with carbodiimides continuously. The diffusive addition of fuel and simultaneous waste removal allows undisturbed microscopic observation of active droplets under non-equilibrium steady-state conditions. Here, we observed a thermodynamically unfavored and size-dependent transition of spherical droplets into spherical shells —a dissipative structure that theory and experiment verified to originate from reaction-diffusion gradients. The demonstrated setup will contribute to observing and elucidating other non-equilibrium behaviors of active droplets, like the control over droplet size and number. The gained insights can help to further our understanding of biological structure formation and how reaction-diffusion gradients can regulate these under non-equilibrium steady-state conditions.

## Methods
### Materials
All solvents were purchased in analytical grade from Sigma Aldrich and used without further purification. N,N-dimethyl formamide (DMF, 99.8%), Fmoc-R(Pbf)-OH (≥99.0%), Fmoc-D(OtBu)-OH (≥98.0%), Fmoc-G-OH (≥98.0%), Fmoc−N(Trt)-OH) (≥97.0%), N,N′-diisopropylcarbodiimide (DIC, 99%), ethyl cyano(hydroxyimino)acetate (Oxyma, Novabiochem©, ≥98.0%), Wang resins (100−200 mesh, 0.4−0.8 mmol g⁻¹), Rink Amide resins (100−200 mesh, 0.4−0.8 mmol g⁻¹), 4-(dimethylamino)-pyridine (DMAP, ≥98.0%), trifluoroacetic acid (TFA, 99%), piperidine (99%), triisopropylsilane (TIPS, ≥98.0%), N,N′-diisopropylethylamine (DIPEA, ≥99.0%), 4-chloro-7-nitrobenzofurazan (NBD-Cl, 98%), polystyrene sulfonate (pSS, 75 kg mol⁻¹, 18 wt% in water), polystyrene sulfonate (pSS, 32 kg mol⁻¹), polystyrene sulfonate (pSS, 151 kg mol⁻¹), sulforhodamine B (75%), 4-morpholineethanesulfonic acid (MES) buffer, carboxylate-modified polystyrene beads (fluorescent yellow-green, 1 μm mean particle size), Cy3-labeled (ATCG)₈, and polyuridylic acid potassium salt (pU) were all purchased from Sigma-Aldrich and used without any further purification unless indicated otherwise. Peptides Ac-F(RG)₃D-OH (99%) and Ac-F(RG)₃N-NH₂ (99%) were purchased from CASLO Aps. Peptides NBD-G(RG)₃N-NH₂ and NBD-G(RG)₃D-OH were synthesized using a published procedure[23]. Cy5-pSS (710 kg mol⁻¹) was synthesized using a published procedure[54]. Cy3-U₁₅ was purchased from biomers.net GmbH. Cy3-dextran sulfate (40 kg mol⁻¹) was purchased from CD Bioparticles. 2 w% 008-Fluorosurfactant in Novec 7500 was purchased from

RAN Biotechnologies. Novec 7500 was purchased by Iolitec. Poly(-vinylsulfonic acid) sodium salt (pVS, 25 w% in water) was purchased from Polysciences.

## Standard sample preparation

Stock solutions of Ac-F(RG)$_3$-D-OH (precursor, 300 mM), pSS (41 mM, according to monomer units, 7.6 mg mL$^{-1}$), polyuridylic acid (41 mM, according to monomer units, 13.3 mg mL$^{-1}$), polyvinyl sulfonate (100 mM, according to monomer units, 10.8 mg mL$^{-1}$) and MES (650 mM) were prepared by dissolving the respective amount of each component in MQ water and adjusting the pH to 5.3. Stock solutions for Ac-F(RG)$_3$N-NH$_2$ (product*) and the fluorescent dyes were also prepared in MQ water but without pH adjustment. All stocks were filtrated with a syringe filter (PTFE, 0.2 μm pore size). For active droplets fueled with a batch of fuel: 5–20 mM DIC (fuel) was added to a solution (20–500 μL) of precursor and pSS in 200 mM MES at pH 5.3, and the solution was mixed with a pipette. For active droplets fueled with excess fuel: 5 μL of DIC is added to a solution (20 μL) of 10 mM precursor and 5 mM pSS in 200 mM MES at pH 5.3. The sample is mixed with a pipette, and residual DIC above the solubility limit is left as a reservoir on top of the aqueous solution. For active droplets fueled continuously, a solution (5 μL) containing precursor and pSS in 200 mM MES at pH 5.3 was mixed with perfluorinated oil (Novec 7500, 50 μL) containing the fluorosurfactant and 0.025–1.0 M DIC. For passive droplets that are not fueled, pSS was added to a solution containing precursor and product* in 200 mM MES at pH 5.3, and the solution was mixed with a pipette.

## Nuclear magnetic resonance spectroscopy (NMR)

$^1$H-NMR spectra were recorded with 16 or 64 scans at room temperature on Bruker AVHD 300, AVHD400, and AVHD500 spectrometers. Chemical shifts are reported in parts per million (ppm) relative to the signal of the deuterated solvent D$_2$O ($\partial$ = 4.7 ppm). All measurements were performed in triplicate ($N$ = 3) at 25 °C. Concentrations were determined from the peak integrals, which were compared with the acetonitrile reference. The chemical shifts of the compared signals were $^1$H NMR (300 MHz, D$_2$O): ACN $\delta$ (ppm) = 2.07 (s, 3H, CH$_3$), DIC $\delta$ (ppm) = 1.22-1.23 (d, 12H, CH$_3$), DIU $\delta$ (ppm) = 1.09–1.10 (d, 12H, CH$_3$).

## High-performance liquid chromatography (HPLC)

High-pressure liquid chromatography was performed using analytical HPLC (ThermoFisher, Vanquish Duo UHPLC, and Thermofisher Dionex Ultimate 3000) with a Hypersil Gold C18 column (100 mm × 3 mm, 250 mm × 4.6 mm). Separation was performed using a linear gradient of acetonitrile (2% to 98%) and water with 0.1 vol% TFA, and the chromatogram was analyzed using detectors at 220 nm and 254 nm. The data was collected and analyzed with the Chromeleon 7 Chromatography Data System Software (Version 7.2 SR4). All measurements were performed in triplicate ($N$ = 3) at 25 °C.

## Fluorescence spectroscopy

Fluorescence spectroscopy was performed on a Jasco spectrofluorimeter (Jasco FP-8300, SpectraManager software 2.13) with external temperature control (Jasco MCB-100). All experiments were performed in triplicate ($N$ = 3) at 25 °C.

## Isothermal titration calorimetry (ITC)

ITC experiments were performed with a MicroCal PEAQ-ITC from Malvern Pananalytical. All experiments were performed at 25 °C. The following conditions were used: pSS (75 kg mol$^{-1}$, 1 mM sulfonate units in MES 200 mM, pH 5.3) was titrated with the precursor (15 mM in 200 mM MES, pH 5.3): 25 injections of 1.5 μL each. pSS (75 kg mol$^{-1}$, 0.05 mM sulfonate unit in 200 mM MES at pH 5.3) was titrated with the product* (1.5 mM as charged units, 0.5 mM as molecular concentrations in 200 mM MES, pH 5.3): 25 injections of 1.5 μL each. All

experiments were performed in triplicate ($N$ = 3). For each experiment, a control was performed by titrating the corresponding amount of peptide (precursor or product*) in 200 mM MES buffer (pH 5.3) without pSS. Data were fitted using a non-linear least squares algorithm provided with the PEAQ-ITC Analysis software.

## Kinetic model

The concentration profiles of the precursor and its corresponding anhydride (product) were determined by HPLC, whereas NMR quantified fuel and waste. Due to the instability of the product, a quenching technique was used with an amine that converts the product into a quantifiable amide[18,23,61]. To a solution (145 μL) of 10 mM precursor in 200 mM MES at pH 5.3 was added a solution (5 μL) of 5–20 mM DIC in acetonitrile in an HPLC vial. After each time point, 10 μL of the reaction mixture was added to 20 μL of an aqueous solution of ethylamine (400 mM). The resulting clear solution was then injected into the HPLC, and the concentration of the precursor and product was calculated from the resulting peak integrals. To quantify DIC and DIU, 5–20 mM DIC was added to a solution (1 mL) of 200 mM MES at pH 5.3 and vortexed for 30 s to dissolve all DIC. The sample also contained 10 vol% D$_2$O and 80 mM acetonitrile (ACN) as a ref. 10 mM precursor was added to 500 μL of this solution to start the reaction network. The concentrations were determined by $^1$H-NMR spectroscopy. The resulting experimental data were fitted to a custom program in Python 3 previously published by the group of Hartley and applied to similar systems[62]. Further information can be found in the Supplementary Methods.

## Method to continuously fuel active droplets

Surfactant-stabilized water in oil droplets was produced using 1 w% 008-FluoroSurfactant in 3 M HFE7500 as the oil phase. To form microreactors of varying size, 5 μL of a solution containing 10 mM precursor and 5 mM pSS in 200 mM MES at pH 5.3 were added to 50 μL of the oil phase in a 200 μL Eppendorf tube. Active droplets: For the preparation of active droplets, fuel was added to the oil phase before adding the sample containing the precursor and pSS. Snipping of the centrifugal tube resulted in the formation of microreactors with a random size distribution. The microreactors were imaged at the confocal microscope in untreated observation chambers[63] consisting of a 24 mm × 60 mm glass cover slide and a 16 mm × 16 mm glass cover slide that were separated by two slices of double-sided sticky tape and sealed with two-component glue. Passive droplets: For the preparation of passive droplets, coacervation is induced right before the encapsulation into the microreactors by adding pSS as the last component. The microreactors were sealed and imaged in untreated observation chambers. Dissolution and reinduction: To exchange the oil phase, 30 μL of a solution containing 10 mM precursor and 5 mM pSS in 200 mM MES at pH 5.3 were added to 300 μL of the oil phase containing the fuel in a 1 mL Eppendorf tube. After 2 h, a sample was imaged in an observation chamber as a control for the formation of spherical shells. The residual microreactors in the Eppendorf tube were pipetted into 300 μL of perfluorinated oil containing no fuel. A sample of these microreactors was imaged in an observation chamber to confirm the dissolution of active droplets and shells. Again after 2 h, the residual microreactors in the oil containing no fuel were pipetted into 50 μL of perfluorinated oil containing 0.5 M DIC. A sample of these microreactors was imaged in an observation chamber.

## Confocal fluorescence microscopy and droplet analysis

A lightning SP8 confocal microscope (Leica) with a 63x water immersion objective (1.2 NA) was used to analyze the coacervates in bulk and the microfluidic reactors (microreactors). Sulforhodamine B was added to track the coacervates via fluorescence if not indicated otherwise, and the dye was excited at 552 nm and imaged at

565-700 nm with a HyD detector. The pinhole was set to 1 Airy unit. Coacervate-based droplets and coacervate-based spherical shells were analyzed with ImageJ (Fiji). For the analysis, images of coacervate-based droplets were thresholded with the moments thresholding algorithm, and a Gaussian blur was applied. Subsequently, the images were analyzed with the analyze particle plugin in ImageJ. The used parameters were size = 5 $\mu m^2$-infinity, circularity = 0−1, and all images were analyzed once with holes included and once without holes. The thickness of the coacervate-based spherical shells was determined by subtracting the vacuole's radius from the droplet's radius with holes. The volume of coacervate-based spherical shells was calculated assuming a spherical shape and considering the half-sphere shape due to wetting at the interface of the microreactor and the oil. The volume of coacervate-based droplets without vacuoles was calculated assuming a spherical droplet.

### Fluorescence recovery after photobleaching (FRAP)
The diffusivity of the molecules inside active droplets was measured via spot bleaching. Measurements were performed in microfluidic reactors. For the recovery of the peptides, coacervate-based droplets were stained with NBD-G(RG)$_3$D-OH (fluorescent analog of precursor) or NBD-G(RG)$_3$N-NH$_2$ (fluorescent analog of product*), which was excited at 488 nm and imaged at 565−635 nm with a PMT detector. For the recovery of pSS, Cy5-pSS was excited at 638 and imaged at 648−720 nm with a PMT detector. A spot size of $r = 1\,\mu m$ in radius was bleached in all experiments. Recovery data were normalized through double normalization[64] and fitted to a first-order exponential equation[65] to obtain the diffusion coefficient.

### DLS analysis
DLS analysis was performed on a Litesizer 500 particle size analyzer equipped with a 658 nm 40 mW laser diode. Samples were transferred into an Omega cuvette (Anton Paar) for $\zeta$-potential measurements. The Smoluchowski approximation was used for $\zeta$-potential analysis. Static droplets composed of 9.5 mM precursor, 0.5 mM product*, and 5 mM pSS in 200 mM MES at pH 5.3 were used to mimic the peptide and anhydride concentrations under steady-state conditions. Static samples were diluted 10-fold with MQ water right before analysis after incubation for 10 min.

### Determination of the partitioning and concentration of molecules inside and outside the droplet phase
We used a centrifugation assay to determine the amount of precursor, product*, and fuel in the dilute phase and inside passive droplets for the phase diagram. For the precursor, we quantified the fraction of the precursor that remained in the dilute phase of passive droplets by HPLC. Samples of passive droplets (150 µL) were incubated for 5 min, and then centrifuged for 15 min at 20,412 × $g$. 100 µL of the supernatant was removed and added into an HPLC inlet. 1 µL of NaCl (4 M) aqueous solution was added to dissolve residual turbidity. The resulting clear solution was injected into the HPLC and compared to a sample with the same conditions but without pSS to calculate the fraction of precursor that remained in the supernatant after droplet formation. For the product, we quantified the fraction of the product that remained in the dilute phase of passive droplets by fluorescence spectroscopy because its concentration in the supernatant could not be determined accurately via HPLC (peak overlap with the precursor and the low fraction remaining in the supernatant). Therefore, we used NBD-G(RG)$_3$N-NH$_2$, which partitions similarly into the droplet phase as product*[54]. Samples of passive droplets (150 µL) containing 1 µM NBD-G(RG)$_3$N-NH$_2$ were incubated for 5 min and then centrifuged for 15 min at 20,412 × $g$. 100 µL of the supernatant was removed and added into a quartz cuvette. 1 µL of NaCl (4 M) aqueous solution was added to dissolve residual turbidity. The fluorescence of the sample was then measured on the fluorimeter (Excitation at 467 nm, emission at 526 nm). To account for

the dependence of the fluorescence of dyes on their environment[66], we prepared an identical sample without NBD-G(RG)$_3$N-NH$_2$. We added the supernatant (96 µL) to a centrifugal tube containing 1 µL of an aqueous solution of NaCl (4 M). To the clear solution, we then added 1 µM NBD-G(RG)$_3$N-NH$_2$ (4 µL) as in the previous sample and measured its fluorescence intensity. The intensity ratio between these two samples was used as the fraction of the fluorescent molecules that remained in the supernatant. We quantified the fraction of fuel partitioned into the droplet phase of passive droplets using NMR. Samples of passive droplets (1 mL) were incubated for 5 min, centrifuged for 15 min at 20,412 × $g$ and the supernatant was carefully removed. The residual coacervate phase (about 1 µL) was dissolved in a solution (50 µL) containing 400 mM NaCl, 560 mM borate buffer at pH 10, 80 mM ACN as standard, and 20 vol% D$_2$O. The high pH prevents further hydrolysis of DIC between sample preparation and measurement. the DIC concentration was determined by $^1$H-NMR spectroscopy. Blank measurements without pSS and product* (no droplets) were performed to account for residual DIC on the walls of the sample containers after the removal of the supernatant. The blank solution without droplets was treated similarly to the solution with droplets. The total droplet volume was estimated via a centrifugation assay. Samples of passive droplets (150 µL) containing 15 µM sulforhodamine B (for visualization) were incubated for 5 min, and then centrifuged for 15 min at 20,412 × $g$. The volume of the droplet pellets (0.4−2 µL) was then compared to size standards visually. From this data, the partitioning coefficient ($K$) of the individual molecules was calculated (Supplementary Tables 4−6).

### Estimating the concentration of product and fuel in the microfluidic droplet
To validate whether the continuously fueled microreactors contain a steady-state level of product and fuel, we chose an indirect quantification method by HPLC for the product[61], and quantification by NMR for the fuel and waste. We assumed that fuel diffusion from the surrounding oil phase into a microfluidic reactor ($r = 5$−50 µm) is fast and should be comparable to a two-phase system in a glass vial under vigorous stirring. We used perfluorinated oil without surfactant in these experiments to avoid stabilized emulsions. To determine the product concentration, 250 µL of an aqueous solution containing 10 mM precursor in 200 mM MES at pH 5.3 was added on top of perfluorinated oil (Novec 7500, 1 mL) in an HPLC vial, stirred vigorously, and with the consecutive addition of 0.5 M to the oil phase, the reaction network was started. In intervals, 10 µL aliquots were taken from the aqueous phase and quenched with 20 µL of an aqueous solution of ethylamine (400 mM). The reaction between the amine and the product yields a stable amide that can be quantified[18,61]. We correlated the resulting peak integrals of the amide to the product, assuming that the absorption of the amide is equal to the absorption of the precursor. To determine the fuel and waste concentration, 1.25 mL of 10 mM precursor in 200 mM MES at pH 5.3 was added to 5 mL of perfluorinated oil (Novec 7500) containing 0.5 M DIC. The mixture was vigorously stirred, and samples of 25 µL were taken and quenched with 25 µL 640 mM borate buffer at pH 10 with 20 vol% D$_2$O and 80 mM acetonitrile as a reference. The high pH inhibits the direct DIC hydration and the reaction of the precursor with DIC to ensure that the time until the sample is measured does not influence the measured concentrations of fuel and waste. Samples are measured by $^1$H-NMR spectroscopy. To determine the fuel concentration in the microreactors depending on the fuel concentration in the oil phase, 1 mL of 200 mM MES at pH 5.3 is added to 10 mL of perfluorinated oil (Novec 7500) containing 0.25−1.0 M DIC. The mixtures were vortexed for 20 s and after the phases had separated, 250 µL of the aqueous phase was added to 250 µL of 640 mM borate buffer at pH 10 with 20 vol% D$_2$O and 80 mM acetonitrile as a reference. Concentrations were determined by $^1$H-NMR spectroscopy.

## Theoretical model

In the theoretical model, we use an effective ternary mixture composed of a solvent component, precursor A, and product B. The influence of fuel, waste, and polyanion is accounted for implicitly. We determined the molecular interaction parameters of a ternary Flory-Huggins free energy density from experimentally measured equilibrium concentrations of the precursor and product. Furthermore, experimentally determined values of kinetic parameters, such as reaction rates and diffusivities, were used in linearized reaction-diffusion equations for obtaining spatial concentration profiles. We solved these kinetic equations by assuming local phase equilibria at a sharp interface in two spherically symmetric geometries. First, we considered an active droplet at the center. Once the concentrations at the core of this droplet lie within the spinodal regime of the phase diagram, the system is locally unstable. Therefore, we also solved a second geometry with a spherically symmetric shell of the dense phase that encloses a more dilute inner phase while surrounded by another dilute phase. Thus, two interfaces exist. Further information on the calculations can be found in the Supplementary Methods.

## Statistics

Statistical analyses were performed using Excel. Experimental data were analyzed using a two-tailed $t$-test for two samples assuming unequal variances. $P < 0.05$ was considered statistically significant. The exact sample size $n$, alpha level $\alpha$, and $P$-value for each test are given in the source data or Supplementary Information.

## Reporting summary

Further information on research design is available in the Nature Portfolio Reporting Summary linked to this article.

# Data availability

The data that support the findings of this study are available from the corresponding authors upon request. Source data are provided with this paper.

# Code availability

The code used to generate the data is available from the corresponding authors upon request.

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

## Acknowledgements

The BoekhovenLab is grateful for support from the TUM Innovation Network - RISE funded through the Excellence Strategy and the European Research Council (ERC starting grant 852187). This research was conducted within the Max Planck School Matter to Life, supported by the German Federal Ministry of Education and Research (BMBF) in collaboration with the Max Planck Society. Funded by the Deutsche Forschungsgemeinschaft (DFG, German Research Foundation) under Germany's Excellence Strategy - EXC-2094 – 390783311. C.W. acknowledges the European Research Council (ERC) under the European Union's Horizon 2020 research and innovation program (Fuelled Life, Grant Number 949021) for financial support.

## Author contributions

J.Bo. and C.W. designed the project. J.Bo., C.W., and F.J. supervised the project. A.M.B and C.D. designed the experiments. A.M.B., C.D., M.S., and A.H. performed the experiments. A.M.B., C.D., and M.S. analyzed the data. J.Ba. and G.B. performed the theoretical calculations. J.Bo., C.W., A.M.B., J.Ba., G.B., and C.D. wrote and edited the manuscript.

## Funding

## Competing interests

The authors declare no competing interests.
