## [Peer Review File · Nature Communications]

Liquid spherical shells are a non-equilibrium steady state of active dropletsReviewers' Comments:

Reviewer #1:

Remarks to the Author:

The author has adequately addressed the questions I raised in the previous round of review, and I have carefully read the author's responses to the other reviewers' inquiries. I believe this work is highly innovative, well-developed, and has the potential to capture the attention of readers from diverse fields. Therefore, I recommend its publication in its current form.

Reviewer #2:

Remarks to the Author:

The revised manuscript has sufficiently answered all the comments raised by me and I recommend its publication in Nature Communication in the current form.

Reviewer #3:

Remarks to the Author:

The authors have improved their manuscript and removed some of the more egregious overstatements. However, I still have concerns that this is essentially one system that has been examined by confocal microscopy and it is not at all clear to me that this is more than an interesting observation with little real utility or interest beyond this isolated example. The authors explain in their response letter that this system was the "best suited" which could be taken to mean it is the only one that worked reliably. Whether this is worthy of publication by itself in such a journal is a very open question in my mind. However, other referees seemed more enthusiastic so maybe I am missing something.

As specific comments in the revised version,

Why is less fuel used than peptide in the first study (20 mM compared to 22 mM)? Is this simply a solubility limit (if I read extended Fig 3D correctly)? The authors make a statement later that "Surprisingly, large reactors produced stable droplets beyond a steady-state fuel concentration of 20 mM." around Figure 4F. Again, if I read this correctly, there is no experimental data above (about) 17 mM and it is not clear why. What happens if the peptide concentration is dropped to 20 mM? There are aspects like this which are not expanded upon but raise questions as to exactly how big the phase space over which this effect exists.

Similarly, does the molecular weight of the PSS used matter? Does the [PSS] matter? Does the polyanion have to be PSS for this effect to work?

As a minor point, it is not totally clear what "41 mM, according to monomer units" means. In the experimental, it would probably best to include absolute masses added.

Have the authors checked whether the dye used for the confocal imaging has any impact on the process? I realise that this is unlikely, but there is risk that this is impacting the behavior in some way.

In extended Fig. 2, it would be useful to match up images in A and B as opposed to showing different time points.

In the experimental for the DLS, the authors use 0.5 mM product*. What is product*?

Reviewer #4:

Remarks to the Author:

Although the original submission to Nature Chemistry was somewhat over-sold, I think the revised paper has sufficient scientific originality and general interest to deserve publication in Nature Communications. Some of the more outlandish claims of biological importance have been toned down, and overall the presentation is now balanced and reasonable (apart from excessive use of words like 'exciting' to describe the results). The observations of self-assembled shells in this system, which while chemically specific is probably representative of a larger class of cases where the same mechanism could operate, are worthy of a wide audience. While the authors admit in effect to using the same data to construct their model as to test it, enough data is involved and the resulting picture sufficiently coherent, that the model does offer valuable explanatory power (if not predictive power).

There is one minor quibble that I did not pick up on previously, which is the use of the word 'efficiency' in relation to the 72% ratio between the energy flux out of the vesicle zone and the global input from the reactions. I don't think the first of these can be considered 'work' -- especially, one cannot talk of the 'work' to sustain the nonequilibrium interfacial structure since the interfacial energy is an energy not a power. The ratio is interesting, but needs a different name -- particularly as the 72% figure would be very high if it actually was an efficiency (at least in any quasi-biological setting).

Response to reviewers for **Liquid spherical shells are a non-equilibrium steady state**

REVIEWER COMMENTS

Reviewer #1 (Remarks to the Author):

The author has adequately addressed the questions I raised in the previous round of review, and I have carefully read the author's responses to the other reviewers' inquiries. I believe this work is highly innovative, well-developed and has the potential to capture the attention of readers from diverse fields. Therefore, I recommend its publication in its current form.

Thank you for your time invested in our work.

Reviewer #2 (Remarks to the Author):

The revised manuscript has sufficiently answered all the comments raised by me and I recommend its publication in *Nature Communication* in the current form.

Thank you for your time invested in our work.

Reviewer #3 (Remarks to the Author):

The authors have improved their manuscript and removed some of the more egregious overstatements. However, I still have concerns that this is essentially one system that has been examined by confocal microscopy and it is not at all clear to me that this is more than an interesting observation with little real utility or interest beyond this isolated example. The authors explain in their response letter that this system was the "best suited" which could be taken to mean it is the only one that worked reliably. Whether this is worthy of publication by itself in such a journal is a very open question in my mind. However, other referees seemed more enthusiastic so maybe I am missing something.

As specific comments in the revised version,

1. Why is less fuel used than peptide in the first study (20 mM compared to 22 mM)? Is this simply a solubility limit (if I read extended Fig 3D correctly)? The authors make a statement later that "Surprisingly, large reactors produced stable droplets beyond a steady-state fuel concentration of 20 mM." around Figure 4F. Again, if I read this correctly, there is no experimental data above (about) 17 mM and it is not clear why. What happens if the peptide concentration is dropped to 20 mM?

Thank you for these remarks. We should have clarified the change in the conditions more clearly. We only added 20 mM of DIC in the first study because this is the determined solubility limit of DIC in the aqueous solution. Adding further DIC forms a DIC oil layer on top of the sample. Similarly, the steady-state concentration of DIC cannot be increased above 20mM in the aqueous reactors. The experimental data under steady-state conditions is only shown for fuel concentrations up to 17 mM. Adding more DIC in the perfluorinated oil to obtain more than 17mM to partition in the aqueous solution would change the perfluorinated such that no stable emulsion is obtained. We clarified that in the manuscript: "Fuel concentrations above 16.8 mM could not be tested experimentally because the necessary high fuel concentrations in the oil phase did not yield stable reactors."

The precursor and pSS concentrations in the first study for batch fueling are higher to increase the size of the obtained droplets. Due to the higher overall abundance of fuel and the spherical geometry of the reactors, sufficiently large coacervate droplets could be obtained under steady-state conditions for reduced precursor and pSS concentrations. We chose these conditions to minimize fuel gradients within the reactors. We clarified this in the manuscript: "Notably, we reduced both the precursor (10 mM) and pSS (5 mM) concentrations in comparison to the batch-fueling experiments to minimize fuel gradients within the reactors."

2. There are aspects like this which are not expanded upon but raise questions as to exactly how big the phase space over which this effect exists. Similarly, does the molecular weight of the PSS used matter? Does the [PSS] matter? Does the polyanion have to be PSS for this effect to work?

Thank you for these suggestions. We agree that the phase space upon which spherical shell formation can be observed is a valuable addition to the work. We added to the manuscript: "We further tested the robustness of the observed shell formation. We found that shells were formed for varying pSS and precursor concentrations and different pSS chain lengths. However, changing the polyanion for polyvinyl sulfonate or polyuridylic acid did not lead to the formation of shells (supplementary discussion 3)." We further clarified in the supporting information: "To test the robustness of the spherical shell formation, we tested different types of poly anions, different chain lengths of pSS, different pSS and precursor concentrations, and the influence of the dye (Supplementary Fig. 13A-D). For the polyanion polyvinyl sulfonate (pVS), no spherical shell formation was observed. We hypothesize that this is the case because no droplets larger than Vunstable were formed (Supplementary Fig. 13E). For the polyanion polyuridylic acid (pU), no spherical shells were observed either, even though droplets larger than

Vunstable were obtained (Supplementary Fig. 13F). Most likely, the previously observed meta-stability of pU-based droplets plays a significant role in the prevention of spherical shell formation.² Under the screened conditions, only pSS-based droplets grew sufficiently large while preserving their fuel-responsive nature to form spherical shells. However, we observed no evidence that other polyanions that yield coacervate-based droplets with similar properties to pSS-based droplets could not form spherical shells.

The reduction of the pSS chain length from 75kDa to 32kDa had no significant effect on the spherical shell formation (Supplementary Fig. 13G). However, no spherical shells could be obtained upon increasing the chain length of pSS to 150kDa due to the formation of droplets prior to the addition of DIC.

Similarly, a reduction in the pSS concentration led to the formation of droplets prior to the addition of DIC (Supplementary Fig. 13H). For pSS concentrations of 5 and 7 mM, spherical shell formation was observed, showing no significant difference in the formation (Supplementary Fig. 13I and J). For 10 mM pSS, only small droplets were formed that did not fuse into one droplet even after 5 h (Supplementary Fig. 13K), preventing proper analysis of spherical shell formation.”

Lastly, we used the labeled precursor NBD-G(RG)₃D-OH (Supplementary Fig. 13L), which showed no significant difference in the spherical shell formation in comparison to sulforhodamine B, verifying that the dye has no significant effect on the formation of the shells.” The additional data is shown in Supplementary Figure 13.

3. As a minor point, it is not totally clear what “41 mM, according to monomer units” means. In the experimental, it would probably best to include absolute masses added.

Thank you for pointing this out. We added the absolute masses to the methods section.

4. Have the authors checked whether the dye used for the confocal imaging has any impact on the process? I realise that this is unlikely, but there is risk that this is impacting the behavior in some way.

We did not observe a significant effect of the dye on the spherical shell formation. See answer to Question 2.

5. In extended Fig. 2, it would be useful to match up images in A and B as opposed to showing different time points.

We chose to show different time points because the experiments in A and B were conducted under different fuel concentrations and show, therefore, a different time evolution, but we agree that this can lead to confusion and clarified the difference in the conditions in the figure.

6. In the experimental for the DLS, the authors use 0.5 mM product*. What is product*?

Product* is Ac-F(RG)₃N-NH₂, which we use as an anhydride analog that is stable against hydrolysis. We added the explanation under standard sample preparation and expanded Supplementary Discussion 1: “Furthermore, product* was also used as a substitute for the product in the formation of static droplets, i.e., droplets whose formation is not governed by chemical reactions.”

Reviewer #4 (Remarks to the Author):

Although the original submission to Nature Chemistry was somewhat over-sold, I think the revised paper has sufficient scientific originality and general interest to deserve publication in Nature Communications. Some of the more outlandish claims of biological importance have been toned down, and overall the presentation is now balanced and reasonable (apart from excessive use of words like 'exciting' to describe the results). The observations of self-assembled shells in this system, which while chemically specific is probably representative of a larger class of cases where the same mechanism could operate, are worthy of a wide audience. While the authors admit in effect to using the same data to construct their model as to test it, enough data is involved and the resulting picture sufficiently coherent, that the model does offer valuable explanatory power (if not predictive power).

There is one minor quibble that I did not pick up on previously, which is the use of the word 'efficiency' in relation to the 72% ratio between the energy flux out of the vesicle zone and the global input from the reactions. I don't think the first of these can be considered 'work' -- especially, one cannot talk of the 'work' to sustain the nonequilibrium interfacial structure since the interfacial energy is an energy not a power. The ratio is interesting, but needs a different name -- particularly as the 72% figure would be very high if it actually was an efficiency (at least in any quasi-biological setting).

Thank you for your suggestion. We now indicate that this ratio can be thought of as a measure of the pump efficiency. Added to the manuscript: “The ratio between the pump power, J_{int} , and the total supplied chemical power, J_{tot} , is approximately 0.7 and can be thought of as a measure of the pump efficiency. J_{int}/J_{tot} is lower than 1 because the deactivation outside and the re-activation inside the shell do not contribute to the flux through the interface.”

Reviewers' Comments:

Reviewer #3:

Remarks to the Author:

The authors have clarified a number of points and added extra information. I believe that the manuscript is now suitable for publication.

REVIEWERS' COMMENTS

Reviewer #3 (Remarks to the Author):

The authors have clarified a number of points and added extra information. I believe that the manuscript is now suitable for publication.

Thank you for your approval of the manuscript. We are glad we could clarify your final points.